genetics/evolution/developmental biology

developmental transcriptome, differential expression, WGCNA, alternative splicing, butterfly wing patterns

**Authors for correspondence:**
Riddhi Deshmukh
e-mail: riddhimd@ncbs.res.in
Krushnamegh Kunte
e-mail: krushnamegh@ncbs.res.in

# Tissue-specific developmental regulation and isoform usage underlie the role of *doublesex* in sex differentiation and mimicry in *Papilio* swallowtails

Riddhi Deshmukh, Dhanashree Lakhe and Krushnamegh Kunte

National Centre for Biological Sciences, Tata Institute of Fundamental Research, GKVK Campus, Bellary Road, Bengaluru 560065, India

RD, 0000-0002-7634-2029; KK, 0000-0002-3860-6118

Adaptive phenotypes often arise by rewiring existing developmental networks. Co-option of transcription factors in novel contexts has facilitated the evolution of ecologically important adaptations. *doublesex* (*dsx*) governs fundamental sex differentiation during embryonic stages and has been co-opted to regulate diverse secondary sexual dimorphisms during pupal development of holometabolous insects. In *Papilio polytes*, *dsx* regulates female-limited mimetic polymorphism, resulting in mimetic and non-mimetic forms. To understand how a critical gene such as *dsx* regulates novel wing patterns while maintaining its basic function in sex differentiation, we traced its expression through metamorphosis in *P. polytes* using developmental transcriptome data. We found three key *dsx* expression peaks: (i) eggs in pre- and post-oviposition stages; (ii) developing wing discs and body in final larval instar; and (iii) 3-day pupae. We identified potential *dsx* targets using co-expression and differential expression analysis, and found distinct, non-overlapping sets of genes—containing putative *dsx*-binding sites—in developing wings versus abdominal tissue and in mimetic versus non-mimetic individuals. This suggests that *dsx* regulates distinct downstream targets in different tissues and wing colour morphs and has perhaps acquired new, previously unknown targets, for regulating mimetic polymorphism. Additionally, we observed that the three female isoforms of *dsx* were differentially expressed across stages (from eggs to adults) and tissues and differed in their protein structure. This may promote differential protein–protein

interactions for each isoform and facilitate sub-functionalization of *dsx* activity across its isoforms. Our findings suggest that *dsx* employs tissue-specific downstream effectors and partitions its functions across multiple isoforms to regulate primary and secondary sexual dimorphism through insect development.

## 1. Introduction

Evolutionary novelties, which often lead to diversification of life forms, have varied developmental origins. While complex adaptations could result from the emergence of novel genes [1], they could also arise by rewiring existing pathways. The pleiotropic nature of several genes, such as transcription factors, allows co-option of developmental circuits in novel contexts at relatively short timescales [2]. For instance, horns in beetles [3], wings in insects [4,5] and mimicry in butterflies [6–11] have resulted from tissue- and developmental stage-specific co-option of existing genes and pathways. Mutations in regulatory regions, synthesis of different gene products by alternative splicing, and mutations in binding sites that enable interactions with new partners, are a few mechanisms by which transcription factors may acquire functions that lead to novel adaptive phenotypes [12,13].

*doublesex* (*dsx*) is a pleiotropic transcription factor that governs sexual dimorphism in insects at various developmental stages. Apart from governing embryonic sex differentiation, it is often co-opted in developing pupae to govern secondary sexually dimorphic phenotypes in adults, such as sex combs in male *Drosophila* [14], exaggerated horns and mandibles in male beetles [15–17] and caste differentiation in social ants [18]. In *Papilio* butterflies, *dsx* regulates female-limited Batesian mimicry and mimetic polymorphism [8,19]. *Papilio polytes*, through most of its geographical range, exhibits two female forms, the mimetic form *polytes*—which mimics the aposematic *Pachliopta aristolochiae*—and non-mimetic, male-like form *cyrus* [20]. These forms are governed by differential expression of alternative alleles of *dsx*, one of which is contained in an inversion and results in *f. polytes* [8,19]. While the inversion maintains two separate alleles and prevents maladaptive intermediates in *P. polytes*, in *Papilio memnon*, mimetic polymorphism is maintained in the absence of an inversion with two alleles of *dsx* [10]. It appears as if this critical gene has evolved multiple mechanisms to maintain and govern different morphs even in closely related species. Exploring the molecular role of *dsx* in the regulation of sexually dimorphic and polymorphic traits in developing insects in all embryonic and other stages during the metamorphosis will help us understand the extent of its molecular pliability to accommodate novel functions and co-option in different developmental contexts while maintaining its basic function in sex differentiation. However, our understanding in this area is somewhat fragmented: the function of *dsx* in embryonic sex differentiation has been studied very well in a few model insect species [21–32], whereas the co-opted functions have been studied in a larger number of non-model organisms [8,10,15–18]. It is necessary that we bridge this disconnect by comparing the action of pleiotropic genes in embryonic sex differentiation as well as that in late-development regulation of secondary sexual traits in a wide range of model and non-model species.

In this study, we investigate the mechanisms by which *dsx* regulates sexual dimorphism in *P. polytes* by using a developmental transcriptome and quantitative polymerase chain reaction (qPCR) datasets across its metamorphosis from ova and eggs to adults. Previously, only the expression profiles and isoform usage in pre-pupal and pupal stages had been studied in this species [8,19]. Thus, we provide new data on gene expression patterns as well as isoform usage in ova and eggs when sex differentiation occurs, and put this in context of the well-known pattern of expression and isoform usage in late pupal development when different wing colour morphs are laid down. Additionally, we identify potential interacting partners of this gene from fifth instar larvae to pupal stages to help elucidate how gene networks have been tweaked to accommodate the pleotropic functions of *dsx*. This will help generate a more coherent understanding of the action of *dsx* from early sex differentiation to late-stage secondary sexual dimorphism in this emerging model species.

## 2. Materials and methods

### 2.1. Sample collection and sequencing

We sampled mimetic *P. polytes* females and non-mimetic males from a greenhouse population of *P. polytes* at different stages of their life cycle (electronic supplementary material, table S1) and separately preserved the tissues in TRIzol™. While sampling pre-oviposition eggs from females, we cleaned out as much somatic

tissue from around the eggs as feasible. We extracted RNA from these tissues using the chloroform-isopropanol-based extraction method and prepared libraries using TruSeq® RNA Sample Preparation Kit v2. We quantified the libraries using Qubit fluorometric quantification and checked their profile using Bioanalyzer. At the time of library preparation and then sequencing, we took equal RNA concentrations across samples to normalize differences between tissue mass (e.g. wing discs have less tissue mass compared to abdomen). Thus, the differential expression revealed in further analyses reflected true differences in gene expression across tissue and lifestages, and not differences in starting tissue mass. We sequenced the transcriptomes using $2 \times 100$ paired-end sequencing runs on Illumina HiSeq 2500 to obtain nearly 20 million reads per sample (accession numbers for raw data are in the electronic supplementary material, table S2).

## 2.2. Transcriptome assembly, differential expression and weighted gene correlation network analysis

We quality-checked the raw sequencing data, trimmed reads and aligned them to the *P. polytes* reference genome [19] using STAR 2.6 [33]. We obtained raw counts from the alignment using HTSEQ [34], which we normalized to plot gene expression using the edgeR package [35] in R [36]. To determine differential expression between male and female tissues of 3-day pupal stage, which is critical for wing patterning [19], we used genewise negative binomial generalized linear models. We chose an false discovery rate cut-off of less than 0.001 and annotated differentially expressed genes using the *P. polytes* annotation on the National Centre for Biotechnology Information (NCBI; https://www.ncbi.nlm.nih.gov/genome/annotation_euk/Papilio_polytes/100/). To represent these in the form of a heatmap, we used the heatmap3 package in R [37]. We used the weighted gene correlation network analysis (WGCNA) package [38] to obtain modules of genes co-expressed with *dsx* in male wings, female wings, male abdomen and female abdomen separately, using read counts from fifth instar larvae, pre-pupae, 3-day-, 6-day- and 9-day-old pupae. We chose the smallest soft threshold power for which our data showed scale-free topology. These powers were different for each run (ranging from 6 to 13) based on the gene composition and connectivity of each tissue. We used a minimum module size of 30 genes, and in each run, *dsx* did not get sorted with unassigned genes. We used the dynamic tree cut function to merge modules with similar expression using the default parameters. We annotated the genes in the *dsx*-containing module using the *P. polytes* annotation from NCBI. For assembling isoforms of *dsx*, we performed *de-novo* assembly of raw reads using TRINITY [39] on the Galaxy server (https://usegalaxy.org) and used BLAST+ [40,41] to identify individual isoforms in each sample.

## 2.3. Expression and structure prediction of *doublesex* and its isoforms

We used reverse transcriptase-PCR (RT-PCR) and qPCR to test for the presence and expression levels of *dsx* and its isoforms F1, F2, F3 and the male isoform (M), across tissues and developmental stages of *P. polytes* (primer details in the electronic supplementary material, table S3). We used ProtoScript® II First Strand cDNA Synthesis Kit to synthesize cDNA and performed RT-PCR and qPCR using ribosomal protein RPL3 as the internal control. We used PowerUp SYBR Green Master Mix (ThermoFisher), for performing qPCR for samples that tested positive for isoform-specific RT-PCR, with the same primers. The three-dimensional structure of *dsx* largely comprises of loops, making tertiary structure prediction unreliable. Therefore, we predicted secondary structures of *dsx* isoforms F1, F2, F3 and M, using PSIPRED [42,43].

## 2.4. *doublesex*-binding motif discovery

We used Hypergeometric Optimization of Motif EnRichment (HOMER) v4.11 [44] to scan the *P. polytes* reference genome for *dsx*-binding motifs, using the canonical *dsx*-binding sequence from *Drosophila melanogaster* [45]. With an 11 bases long binding motif, we found nearly 4200 loci that had a motif similar to that of *Drosophila dsx*. We screened these using candidates from WGCNA and differential expression and found *dsx*-binding motifs in 246 of 478 genes (electronic supplementary material, table S5).

# 3. Results and discussion

## 3.1. *doublesex* shows multiple activity peaks through metamorphosis in *Papilio polytes*

We tracked *dsx* expression across various tissues during *P. polytes* metamorphosis using whole-transcriptome data and *dsx*-specific RT-PCR. The whole-transcriptome sequencing showed little *dsx* expression in eggs and

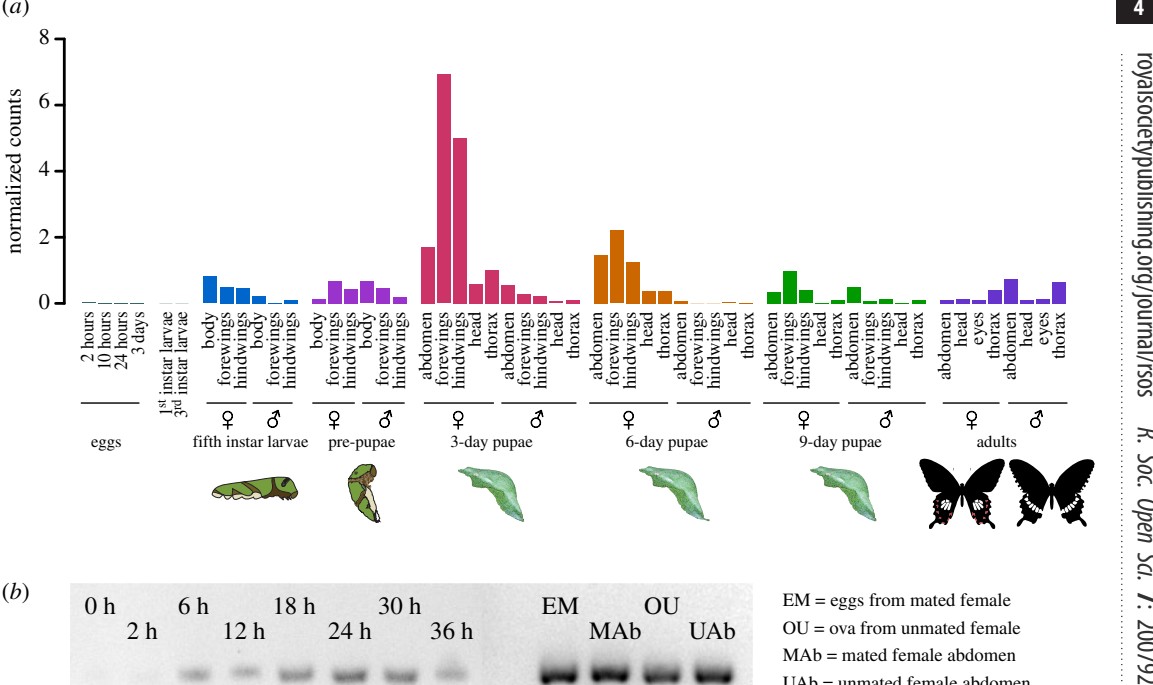

**Figure 1.** Expression of *doublesex* across developmental stages and tissues of *Papilio polytes*. (*a*) Expression of *dsx* from whole-transcriptome data of developing butterflies, from eggs to adults. Here, males represent non-mimetic wing patterns and females represent mimetic wing patterns. (*b*) *dsx* expression in eggs pre- and post-oviposition using RT-PCR. Of the four samples, MAb and UAb corresponded to abdominal tissue of mated (MAb) and unmated females (UAb), excluding eggs and ova. The separated ovaries and eggs from abdominal tissue made up samples of eggs from mated (EM) and ova from unmated females (OU). 0 h eggs were freshly laid, and caterpillars usually hatched after 48 h.

early larval instars compared to wings in pupal stages, suggesting the inherent expression differences at these two stages and tissues (figure 1*a*). However, sampling of eggs at a finer timescale using RT-PCR elucidated the pattern of *dsx* expression in developing eggs before and after oviposition (figure 1*b*). These two methods combined showed two stages with elevated and differential *dsx* activity: (i) egg stage, where ova and eggs before oviposition showed very high expression, and eggs from 6–36 h after oviposition showed lesser but prominent expression levels (figure 1*b*). This is similar to embryonic *dsx* activity in *Bombyx mori* [21], where ovarian eggs exclusively express female isoform of *dsx* (*dsxF*), whereas the male isoform (*dsxM*) starts expressing several hours post-oviposition, both contributing to early sex differentiation [22,25]. The transition period between the expression of female isoform to male isoform might account for variation in *dsx* expression after oviposition. Overall, the embryonic patterns of expression in *P. polytes* are in line with the patterns known in *B. mori*, showing that the basic function of *dsx* in sex differentiation may be conserved across these widely separated species; and (ii) peak *dsx* expression was observed in the developing wings of 3-day-old female pupae (figure 1*a*), as previously reported [19], followed by a gradual decrease through pupal development (figure 1*a*). In addition, there was discernable sexually dimorphic expression in wing discs of the final instar caterpillars, where female wing discs and body showed marginally higher expression levels compared to males (figure 1*a*). Taken together, it is possible that relatively low expression of *dsx* is sufficient for sex differentiation at the early egg developmental stage, but significant upregulation is perhaps required to lay down wing pattern differences in pupal stages. Moreover, it appears that the regulation of wing pattern polymorphism by *dsx* initiates in wing discs of the final larval instar and peaks at 3 days after pupation before slowly trailing off, but maintaining the female-biased upregulation throughout (figure 1*a*).

## 3.2. *doublesex* targets different genes in wings and abdomen of mimetic and non-mimetic wing forms

We identified potential *dsx* targets in tissues with high *dsx* activity, such as developing wings and abdomen in all stages where sexes could be distinguished (fifth instar larvae onwards). We used WGCNA to identify

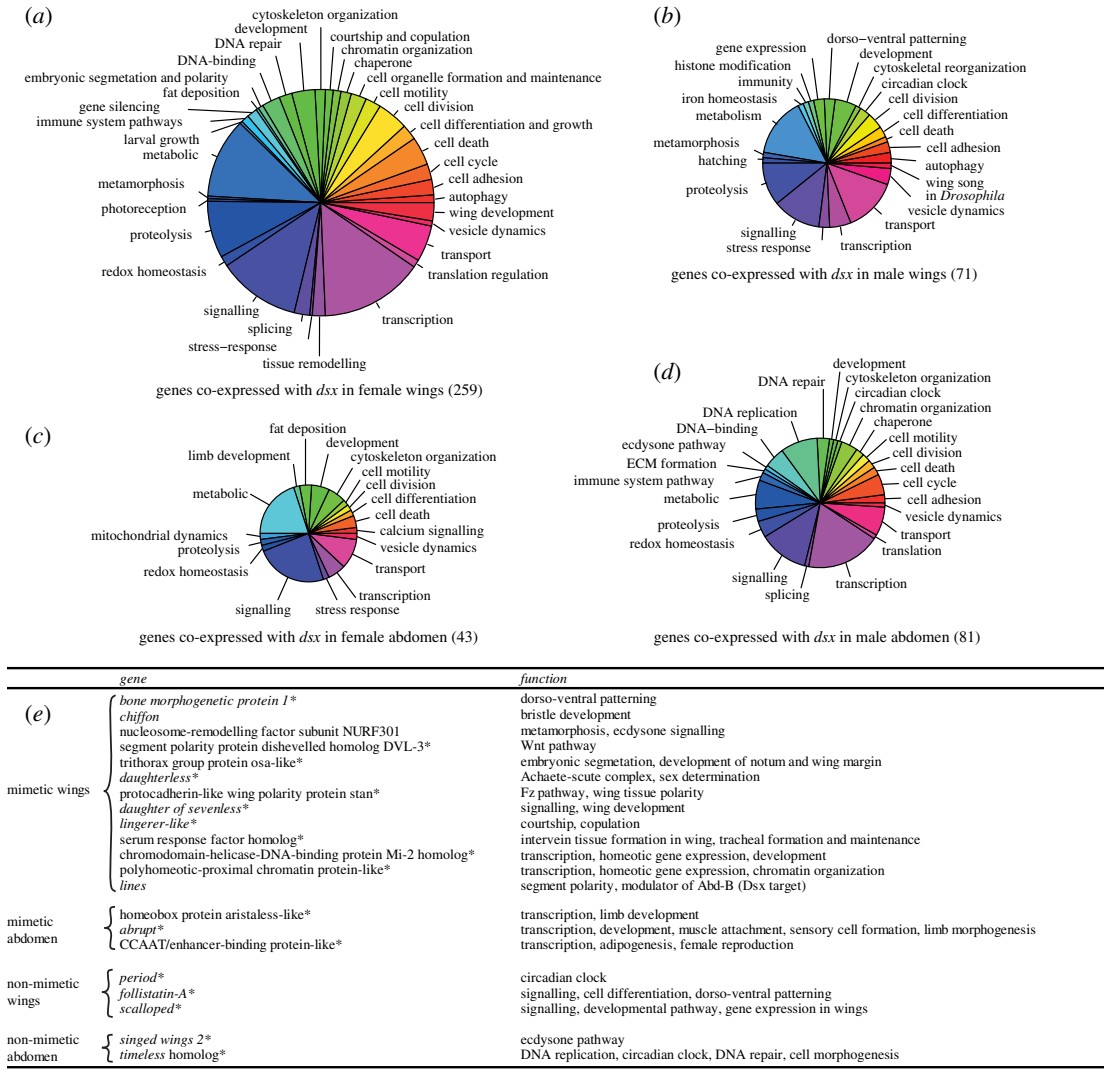

**Figure 2.** Putative targets of *dsx* in wings and abdominal tissue from co-expression analysis using WGCNA. Pie charts (*a*)–(*d*) depict functional categories of genes co-expressed with *dsx* in female wings, male wings, female abdomen and male abdomen respectively. Each pie chart is scaled by the number of genes that clustered with *dsx* in the same module. The table (*e*) highlights genes that are relevant in the context of wing development, patterning and reproduction in both mimetic and non-mimetic comparisons. Asterisk (*) denotes the presence of at least one *dsx*-binding motif in that locus.

genes that co-expressed with *dsx* in mimetic and non-mimetic butterflies. This analysis revealed stark differences between the functionally relevant and non-overlapping pools of downstream targets in mimetic and non-mimetic wings. Genes involved in various physiological and metabolic processes co-expressed with *dsx* in each comparison (figure 2 and electronic supplementary material, table S4), including genes that participate in pathways that *dsx* is involved in and genes that are important in wing development and patterning, such as: (i) dishevelled protein DVL-3 homologue, which is involved in the *Wnt* signalling pathway (figure 2). *Wnt* ligands are involved in wing patterning in several butterfly species: *WntA* in nymphalid butterflies [46,47] and *Wnt1* and *Wnt6* in f. *polytes* of *P. polytes* [48]. The activity of *Wnt1* and *Wnt6* is tied to that of *dsx* in f. *polytes* and both genes are downregulated in *dsx* knockdowns; (ii) trithorax protein *osa*-like, a gene that shows female-biased expression in 3-day pupae and its counterpart in *Drosophila* is required for activation of genes such as *Antp*, *Ubx* and *Eve*, as well as for repression of wingless-regulated genes such as *dll*, and *dpp*, in the absence of Wg signal [49,50]; and (iii) *lin*, a regulator of Abd-B in *Drosophila*. However, *lin* was not highly expressed in the 3-day pupal stage, a characteristic of *dsx* expression in mimetic females, so it is perhaps not involved in regulating mimetic wing morph in *P. polytes*. There was no overlap in co-expressed genes between wings and abdominal tissue, implying the existence of discrete *dsx*-associated molecular networks during wing versus genitalia development, with *dsx* potentially regulating different downstream genes in both

contexts. Several tissue-specific genes co-expressed with *dsx*: wing development and patterning genes in mimetic wings, and genes involved in lipid metabolism in mimetic abdomens (figure 2). On the other hand, non-mimetic wings showed few wing patterning genes co-expressing with *dsx*. This may have resulted from low expression of *dsx* in non-mimetic wings compared to mimetic wings leading to spurious co-expression hits. Some genes that co-expressed with *dsx* in mimetic and non-mimetic wings did not show sex-specific expression in the wings. We retained these genes in figure 2 as they may have a more general role to play in wing development. Overall, nearly all the context-specific genes for each comparison also contained at least one *dsx*-binding motif (denoted by an asterisk in figure 2, complete list in the electronic supplementary material, table S5). Thus, *dsx* may have acquired non-canonical targets such as *Wnt* pathway genes [48] in addition to its known targets in mimetic individuals to govern wing pattern polymorphism. However, this needs to be verified by chromatin-binding and protein interaction assays such as ChIP-seq and co-IP in *P. polytes*. We also assessed differentially expressed genes in 3-day pupae (*dsx* expression peak) of non-mimetic and mimetic butterflies to find relevant potential *dsx* targets (electronic supplementary material, figure S1). Among genes that showed high expression in mimetic wings compared to non-mimetic wings, and that also contained *dsx*-binding motifs, *hedgehog* homologue (*desert hedgehog B*) and *hairy* emerged as potential candidates owing to their involvement in morphogenesis and bristle formation, respectively (bristles are equivalent to scales in Lepidoptera; electronic supplementary material, figure S1).

## 3.3. Female isoforms of *doublesex* co-express in most tissues, but differ in expression levels and protein structures

Three female isoforms of *dsx* occur in *P. polytes*, which differ in their extent of usage of exons 3 and 4 (figure 3*a*) [19,51]. We traced stage- and tissue-specific isoform activity using a combination of isoform-specific search in *de-novo* assembled transcriptomes and PCRs using isoform-specific primers (figure 3*b* and electronic supplementary material, table S6). We observed that all three isoforms were expressed across most *dsx*-expressing tissues during development. Although poorly detected using RT-PCR (electronic supplementary material, table S6), qPCR identified measurable F1 isoform activity in several developmental stages and tissues of *P. polytes* females. In most tissues, however, this expression was low compared to that of isoforms F2 and F3 which seemed to contribute most to *dsx* expression (figure 3*b* and electronic supplementary material, table S6). This was also observed in non-mimetic females (*f. cyrus*) in 3-day pupae (peak *dsx* expression). Isoform-specific expression was similar between mimetic and non-mimetic females, while male-specific isoform showed low expression compared to female isoforms in most tissues except abdomen (figure 3*c*). The difference in relative expression in 3-day pupae in panels (*b*) and (*c*) can be attributed to developmental differences at the individual level among pupae sampled at the 3-day stage. The dichotomy in the expression of isoforms may be owing to varying strength of splice sites between exons 3 and 4. Selective mutation or blocking of splice sites might illustrate their role in isoform expression. Alternatively, isoform F1 might perform a subset of *dsx* functions, which may be specific to early embryonic development and pupal abdomens, where it shows higher expression. The secondary structure prediction for *dsx* isoforms showed that they differed at their C-terminal ends (figure 4*b* and electronic supplementary material, figure S2). Isoforms F1 and F3 showed the presence of helices, while isoform F2 lacked a secondary structure in this region. The C-terminal domain of *dsx* contains the dimerization domain and sex-specific region that is important for the recruitment of cofactors for regulatory functions. Moreover, dimerization or protein–protein interactions of *dsx* can influence DNA binding [54]. It is possible that female isoforms regulate different sets of genes and partition *dsx* activity. However, the relevance of these structural differences needs further investigation.

## 3.4. Proposed mechanism of developmental regulation of sex differentiation and secondary sexual traits by *doublesex*

*dsx* is a crucial gene that regulates various aspects of sexual dimorphism, from embryonic sex differentiation to pupal and adult expression of secondary sexual traits. Its modular architecture with individual genic regions evolving under different selection pressures [51] might facilitate maintenance of its core function in sex differentiation while accommodating the evolution of novel ones in later development. How does a highly conserved gene such as *dsx* regulate different phenotypic outcomes with a limited set of developmental tricks? To address this question, we propose a mode of action for

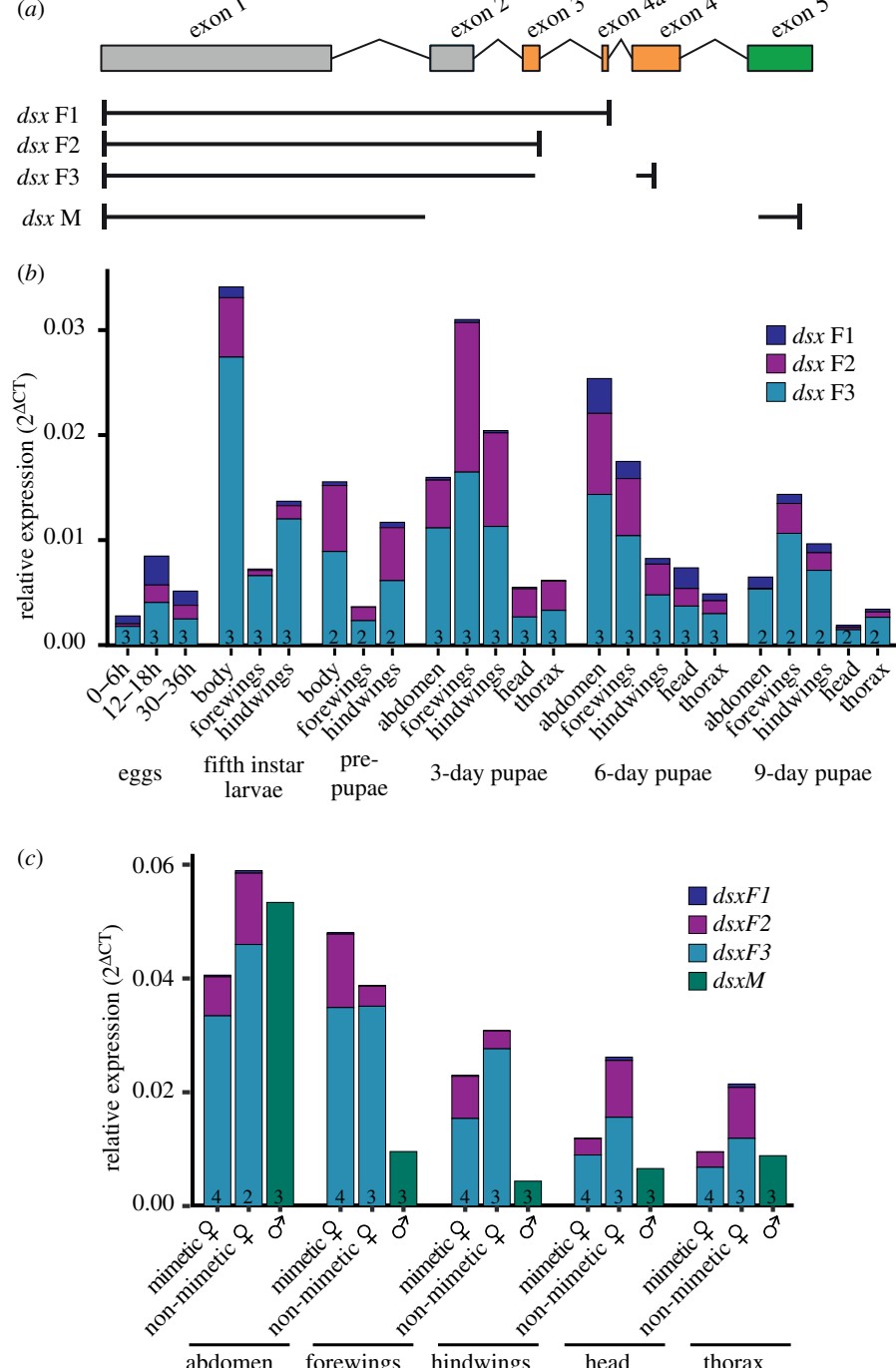

**Figure 3.** Expression of female isoforms of *dsx* in developing *Papilio polytes*. (*a*) *dsx* occurs as three isoforms in *P. polytes* females depending on the usage of female-specific exons 3 and 4. (*b*) Expression of female isoforms of *dsx* using qPCR. The number at the base of the bar indicates sample size for each tissue and stage. The female isoforms of *dsx* show different expression levels across tissues and stages, with F1 usually expressing at lower levels in most tissues compared to F2 and F3, which contribute to most of the *dsx* expression. (*c*) Expression of *dsx* isoforms in mimetic and non-mimetic female forms and males at 3-day pupal stage.

*dsx* in different tissues based on our findings and other studies (figure 4). Figure 4*a* is largely based on what we know from other studies [21,22,25,52], and our hypotheses in *P. polytes* are drawn upon this existing information. Our co-expression analysis in addition to previous work [48] indicates that *dsx* might be governing different targets in mimetic and non-mimetic wings, from which figure 4*b* is drawn. Panel (*c*), and specifically our hypothesis that the three isoforms regulate different, potentially overlapping pools of genes, is based largely on our data on *dsx* isoforms (figure 3 and electronic supplementary material, figure S2). We also take into account the allelic status and tissue specificity of

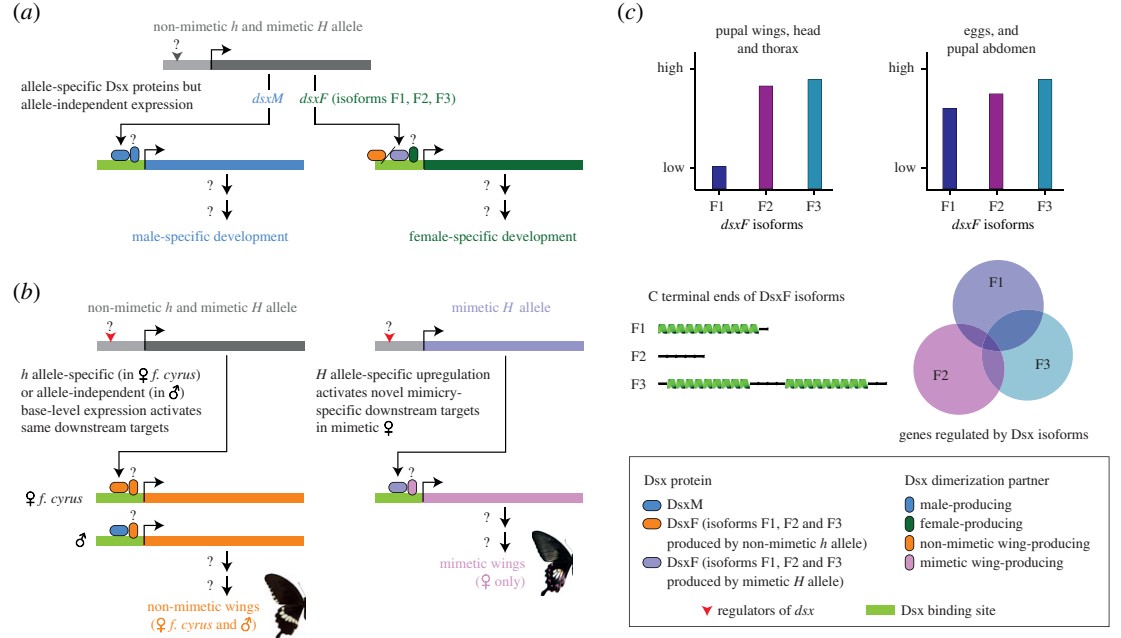

**Figure 4.** Proposed mechanism outlining the role of *dsx* in early-development sex differentiation and novel, late-development wing pattern polymorphism in *Papilio polytes*. (*a*) Early embryonic sex differentiation and late developmental sex-specific reproductive traits (e.g. genitalia and eggs) are highly canalized and strictly sex-specific, irrespective of allelic polymorphism and alternatively spiced isoforms [52,53]. (*b*) Late developmental novel mimetic wing patterns are produced by a combination allelic polymorphism, differential expression of alternatively spiced isoforms, and sex-limitation, expressing mimetic wing patterns only in females that contain at least one copy of the mimetic *H* allele of *dsx* (figure 2, [8,19,48]). (*c*) A graphic representation of relative Dsx isoform activity in regulation of polymorphic wings and other tissues (based on data from [8,19] and figure 3 from this study). Dsx F1 is downregulated in relation to F2 and F3 during the production of secondary sexual traits, but expressed comparably during sex differentiation and basic reproductive traits. The C terminal ends of the three DsxF isoforms are different (based on the electronic supplementary material, figure S2), and their downstream targets are probably different. The specific allelic variants [8,19], sex-specific and alternatively spliced Dsx proteins ([8,19] and the electronic supplementary material, figure S2 in this study), and molecular identities of downstream targets of *dsx* that regulate mimetic wing patterns ([48] and figure 2; electronic supplementary material, figure S1 in this study) have recently been identified. This model has several unexplored aspects that need to be studied in the future, including upstream regulators of *dsx* in each context, precise downstream targets, and both allele- and isoform-specific and protein-interacting partners of *dsx*. Such genes, regulators and networks with as yet unknown molecular genetic identities and mechanisms are marked with '?'.

*dsx* in regulating wing pattern polymorphism and reproductive traits in male and female *P. polytes*, which has not been considered fully before. *doublesex* may have fundamentally different modes of action for early embryonic sex differentiation and late developmental (final instar and pupal) regulation of secondary sexual traits.

The embryonic sex differentiation may be based purely on male-specific (*dsxM*) and female-specific (*dsxF*) isoforms and their downstream targets that set in motion basic sex differences, as seen in other insects [51,55]. In *P. polytes*, mimetic alleles do not seem to affect larval phenotype and development [52], and allele-specific phenotypic changes only start appearing late during pupal stages [53]. Therefore, embryonic sex differentiation may occur irrespective of the allelic status of an individual (figure 4*a*). While the role of *dsx* isoforms in sex-determination has been explored in *B. mori*, we lack such information in *P. polytes*, which has the additional complexity of two *dsx* alleles. It is conceivable that secondary sexual traits in simple sexual dimorphisms (i.e. those without polymorphisms), such as genitalia structures and production of eggs and associated fat bodies, may be regulated similarly by sex-specific isoforms. By regulating similar [56,57] or different downstream genes, *dsxM* and *dsxF* can govern sex-specific developmental cascades establishing basic sexual differentiation, irrespective of multiple *dsxF* isoforms in *P. polytes* (figure 4*a*). However, polymorphic secondary sexual traits might be regulated by *dsx* through more complex developmental processes owing to their sex-limited nature. For example, caste-related polymorphism in social insects, and polymorphic Batesian mimicry in *P. polytes* are female-limited and both governed by *dsx*. The switch between mimetic and non-mimetic

wing phenotypes can be attributed to *dsx* alleles, of which the mimetic allele shows high expression in developing wings early in pupal development [19]. The amino acid substitutions (or possibly *cis*-regulatory mutations) between the two alleles might enable the mimetic allele to regulate different downstream targets in mimetic females compared to the non-mimetic allele in female form *cyrus* and *dsxM* in males [48] irrespective of allelic composition (figure 4*b*). Targeted genetic manipulation in coding and non-coding regions of the *dsx* gene, in addition to chromatin-binding assays can help elucidate the repertoire of targets *dsx* interacts with in mimetic and non-mimetic female forms and identify the corresponding sites in the *dsx* sequence crucial for these interactions.

*dsx* can obtain finer control over the regulation of tissue and developmental stage-specific phenotypes by means of alternative isoforms. Our results showed that isoforms F2 and F3 contributed to most of the *dsx* expression in mimetic wings, whereas eggs and 6-day pupal tissues showed somewhat comparable expression of all three isoforms (figures 3*b*, 4*c*). This suggests that developing eggs and abdominal tissue may employ the entire repertoire of *dsx* regulation by all three isoforms, but in wings, elevated expression of *dsx* isoforms F2 and F3, without F1, might be involved in regulating mimetic phenotype. The occurrence of *dsx* isoforms is not universal across insects: (i) isoform F2 from Lepidoptera shares sequence similarity with *dsxF* isoform in Diptera and Coleoptera, (ii) isoform F3 from Lepidoptera shares sequence similarity only with *dsxF* from Coleoptera, and (iii) isoform F1 appears to be unique to Lepidoptera [51]. It is possible that isoform F2, which is most conserved, plays a larger role in basic sex differentiation, which is universal, while F3—which is shared between insect orders Lepidoptera and Coleoptera with super-diverse wing patterns—might play a larger role in regulating wing patterning, among other sexually dimorphic traits. Similarly, the Lepidoptera-specific F1 might be involved in regulating aspects of early and late sex differentiation that are unique to Lepidoptera. It is yet unknown what might drive this differential regulation of downstream targets of *dsx*. One intriguing possibility is that this differential, possibly non-overlapping regulation of downstream targets of *dsx* is facilitated by distinct C-terminal ends of *dsxF* isoforms, which we are showing here, to our knowledge for the first time (figure 4*c*). This can be experimentally validated with genetic manipulation of the exonic regions unique to each isoform along with isoform-specific ChIP-seq.

The work presented here makes two advances. First, we show that, similar to *B. mori*, *P. polytes* embryos produce sex-specific *dsxF* and *dsxM* isoforms. These sex-specific isoforms presumably have similar functions in early sex differentiation cascades, irrespective of multiple *dsxF* isoforms in *P. polytes*. Our expression analysis also appears to suggest that the low *dsx* expression during early embryonic development may be sufficient for sex differentiation, whereas the very high 3-day pupal expression may be necessary for regulating mimetic polymorphism. Both these hypotheses about the expression levels and isoform functions should be experimentally tested in the future. Second, we demonstrate with WGCNA that *dsx* might employ different downstream effectors in developing wings and abdomens of mimetic and non-mimetic individuals. The acquisition of new *dsx*-binding sites may also help it acquire new targets, such as *Wnt* signalling pathway genes [48]. The added complexity of having multiple female isoforms might assist *dsx* in partitioning some of its functions and enabling tissue-specific regulation. Besides Lepidoptera, Coleoptera and Blattodea have two female-specific exons of *dsx* [51,58]. Therefore, these insect groups may have multiple female isoforms, with the inclusion of one or both female exons. Comparison of developmental transcriptomes across species or even orders might help illustrate how regulation by *dsx* has evolved across different sexually dimorphic traits. Further, morph-specific validation of chromatin binding of *dsx* and identification of its protein co-factors in sexually dimorphic adaptations would help understand the mechanisms by which *dsx* has become the jack of all sexually dimorphic trades.

Data accessibility. The raw RNA seq data are deposited in the NCBI SRA database (BioProject PRJNA634605, Accession nos. SAMN15001929 to SAMN15002046; https://dataview.ncbi.nlm.nih.gov/object/PRJNA634605?reviewer=mf4q72fcfvq67spjf5gl1sduop). Accession details for individual raw transcriptome sequences in the NCBI SRA database are given in the electronic supplementary material, table S2. Primer sequences, isoform sequences, sample details, etc. are provided in the electronic supplementary material, Information.

Authors' contributions. R.D. and K.K. designed research; R.D. generated and analysed transcriptome data and wrote the paper; R.D. generated molecular data and D.L. performed motif discovery; K.K. conceived and supervised the project, and wrote the paper with R.D.

Competing interests. The authors declare no competing interests.

Funding. This work was partially funded by a Ramanujan Fellowship from the Department of Science and Technology, Government of India, and an NCBS Research Grant to K.K., support of the Department of Atomic Energy, Government of India, under project nos. 12-R&D-TFR-5.04-0800 and 12-R&D-TFR-5.04-0900 to TIFR/NCBS, a CSIR Shyama Prasad Mukherjee Fellowship to R.D. and an NCBS student fellowship to D.L.

Acknowledgements. We thank Bhavya Dharmaraaj, Athulya Girish and Tulsamma for maintenance of greenhouse populations of *Papilio polytes*; Vaishali Bhaumik and Dipendra Nath Basu for help with R codes; Sai Guha and Sarvesh Menon for help with WGCNA analysis and NCBS Greenhouse Facility for breeding of butterflies.

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
