## [Reviewer comments · Royal Society Open Science]

Review History

RSOS-200792.R0 (Original submission)

Review form: Reviewer 1

Is the manuscript scientifically sound in its present form?

Yes

Are the interpretations and conclusions justified by the results?

Yes

Is the language acceptable?

Yes

Do you have any ethical concerns with this paper?

No

Have you any concerns about statistical analyses in this paper?

Yes

Recommendation?

Accept with minor revision (please list in comments)

Comments to the Author(s)

This manuscript presents a descriptive analysis of gene expression during the development of a mimetic butterfly, *Papilio polytes*, with a focus on the *dsx* gene. *Dsx* is a known regulator of sex-specific development in insects, and has been previously shown to contribute to sex-limited mimicry in *Papilio* spp. In this report, the authors describe the developmental transcriptome of *P. polytes*; identify several alternative isoforms of *dsx* and quantify their expression in different tissues and at different stages of development using RNA-seq and quantitative PCR; use the developmental transcriptome and motif searches to suggest potential downstream targets of *dsx*; and predict the secondary structure of alternative *dsx* protein isoforms. Although this analysis lacks an experimental or hypothesis-testing component, it provides resources and sets the stage for future experimental analyses, so it is a useful contribution to the field.

In general, the paper is straightforward, and the data are well described. However, I think several issues require clarification or improvement. Some of these relate to potential over-interpretation of the data.

Issues related to correlation network analysis (throughout the paper). This is a major concern of mine. With so few samples, the module structure inferred by this type of analysis is notoriously sensitive to parameter settings. First, the authors need to report their parameters, and show that the modules they infer are at least somewhat robust to parameter settings. Otherwise, the notion of “*dsx*-containing module” has little meaning. Second, I strongly suspect that most of the modular structure comes from the use of very different tissues and widely separated developmental stages. In this sense, the different “*dsx*-containing modules” that the authors report for different tissues/stages may simply reflect differential gene expression between tissues and stages, which mostly has nothing to do with *dsx*. You have different modules for different tissues/stages, and *dsx* has to fall out “somewhere” so of course it comes out in different modules in different samples. So perhaps the fact that *dsx* is associated with different “modules” at different stages/tissues tells you very little about the regulatory relationship between *dsx* and other genes in these “modules”. The *Dsx* binding motif is fairly simple, so the fact that many putative “target” genes have that motif may also not mean very much. I urge the authors to re-examine their network analysis more carefully, to understand where the network structure is really coming from, and whether the allocation of *dsx* to particular modules is robust.

Lines 56-57 “It appears as if this critical gene has evolved multiple mechanisms to maintain and govern different morphs even in closely related species” – This statement, while potentially true, is not directly supported by the data presented in this paper. Frankly, this hypothesis seems to be no more likely after this study than it was before.

dsx expression in unfertilized eggs: can the authors please confirm that the eggs were dissected from ovaries in a way that excluded somatic gonad cells? The detection of *dsx* transcripts by PCR in eggs from unmated females is surprising. This is PCR – even a low amount of contamination from somatic tissues could potentially account for this result.

Lines 134-150. The description of putative *dsx* targets seemed quite confusing to me. First, what is the evidence for describing *dvl-3* or *lin* as “known *dsx* targets”? I don’t know of any direct evidence for that. Second, *Abd-B* is a target of *dsx* in *Drosophila*; that does not necessarily mean that it’s also a *dsx* target in Lepidopterans. I noticed that *Abd-B* did not show up in the abdominal “module” (Figure 2). Please be more careful in distinguishing confirmed facts from hypotheses.

Lines 176-228. The model at the end of the paper is highly speculative and rests on very little data. Some critical parts of this model are still only conjectures that remain to be tested by experiments. There’s nothing wrong with a light dose of interesting speculation at the end of a paper, but please make clear which parts of the model are more solid, and which are speculative.

Lines 22-223 "Besides Lepidoptera, Coleoptera is the only other order that has two female-specific exons of dsx". Actually, the same is true for cockroaches (Blatella).

Figure 1 – what exactly are the units for “normalized counts”?

Figure 3 title – you are really describing isoform expression here, not “activity”

Review form: Reviewer 2

Is the manuscript scientifically sound in its present form?

No

Are the interpretations and conclusions justified by the results?

No

Is the language acceptable?

Yes

Do you have any ethical concerns with this paper?

No

Have you any concerns about statistical analyses in this paper?

No

Recommendation?

Major revision is needed (please make suggestions in comments)

Comments to the Author(s)

Manuscript #: RSOS-200792

Title: "Tissue-specific developmental regulation and isoform usage underlie the role of doublesex in sex-limited polymorphic mimicry in *Papilio swallowtails*"

Authors: Riddhi Dexhmukh et al.

Comments

doublesex (dsx) encodes a transcription factor that generally act as a master regulator for sexual differentiation in insects. Recent studies revealed that dsx has diverse pleiotrophic effects to govern unique sexual dimorphic traits such as horns in beetles, wings in insects, and mimicry in butterflies.

In this study, the authors focused on the mimetic phenotype especially observed in female wings in *Papilio polytes*. In this species, dsx regulates female-limited Batesian mimicry. In an attempt to understand how a critical master regulator dsx controls a novel adaptive phenotype just like the Batesian mimicry while maintaing its inherent function of sexual differentiation, the authors performed a developmental transcriptome analysis to identify potential targets of dsx through metamorphosis.

Overall, this is a nicely done, and several interesting observations are represented and discussed. But I think that this study lacks several data essential for their conclusion like "Isoforms F2 and F3 contributed to most of the dsx expression in mimetic wings" and "elevated expression of dsx isoforms F2 and F3 might be sufficient to give rise to mimetic phenotype. Also, it is unclear what of the findings in this study show novelty as compared with the previously reported findings. Therefore, this manuscript is not suitable for publication in this current form.

Major requirements for revision

1. Figure 1A. If the authors want to argue that the higher expression of *dsx* in the forewings and the hindwings is closely related to the mimetic features, then they should compare the expression profile of *dsx* between mimetic females and non-mimetic females. Why did the authors compare it between mimetic females and non-mimetic "males". Such comparison will just simply emerges the difference in *dsx* expression between females and males, which is not directly involved in the mimetic features observed in wings.
2. Figure 2. As pointed above, if the purpose of this study is to identify the putative targets of *dsx* that may be related to the mimetic phenotype, then the authors should compare the transcriptome data between mimetic females and non-mimetic females. The data presented in Figure 2 does not rule out the possibility that it may simply reflect sexual difference because the data was based on the transcriptomic comparison between females and males.
3. Lines 203-207. The authors said that isoforms F2 and F3 contribute to most of *dsx* expression in mimetic wings and that elevated expression of *dsx* isoforms F2 and F3 might be sufficient to give rise to mimetic phenotype. However, again, if the authors want to say so, then they should perform comparative analysis between mimetic females and non-mimetic females.
4. Overall. I think this study severely lacks novelty and uniqueness. It is unclear for me what of the findings in this study show novelty as compared with the previously studies such as Kunte et al (2014), Iijima et al (2018), and Nishikawa et al (2015). The authors should make clear this point.

Decision letter (RSOS-200792.R0)

Dear Dr Kunte,

The editors assigned to your paper ("Tissue-specific developmental regulation and isoform usage underlie the role of doublesex in sex-limited polymorphic mimicry in *Papilio swallowtails*") have now received comments from reviewers.

Boht reviewers raise significant concerns and a number of points that will require careful consideration. We would like you to revise your paper in accordance with the referee and Associate Editor suggestions which can be found below (not including confidential reports to the Editor). Please note this decision does not guarantee eventual acceptance.

Please submit a copy of your revised paper before 23-Jul-2020. Please note that the revision deadline will expire at 00.00am on this date. If we do not hear from you within this time then it will be assumed that the paper has been withdrawn. In exceptional circumstances, extensions may be possible if agreed with the Editorial Office in advance. We do not allow multiple rounds of revision so we urge you to make every effort to fully address all of the comments at this stage. If deemed necessary by the Editors, your manuscript will be sent back to one or more of the original reviewers for assessment. If the original reviewers are not available, we may invite new reviewers.

- Data accessibility

If you wish to submit your supporting data or code to Dryad (<http://datadryad.org/>), or modify your current submission to dryad, please use the following link:
<http://datadryad.org/submit?journalID=RSOS&manu=RSOS-200792>

- Competing interests

- Authors' contributions

- Acknowledgements

- Funding statement

on behalf of Dr Andy Greenfield (Associate Editor) and Steve Brown (Subject Editor)
openscience@royalsociety.org

Associate Editor's comments (Dr Andy Greenfield):

Comments to the Author:

Your manuscript has now received two expert reviews. You will see that they have both concerns, some of which are major. In particular, you should address, if possible, the following points: i) ensure that the expression profile comparisons are appropriate; ii) re-examine your network analysis more carefully; iii) make sure there is no over-interpretation of the data and clarify its novelty. If you think you are able to address these concerns, your manuscript will be re-considered by the external reviewers. This is not a provisional guarantee that your manuscript will be found acceptable for publication.

Comments to Author:

Reviewers' Comments to Author:

Reviewer: 1

Comments to the Author(s)

This manuscript presents a descriptive analysis of gene expression during the development of a mimetic butterfly, *Papilio polytes*, with a focus on the *dsx* gene. *Dsx* is a known regulator of sex-specific development in insects, and has been previously shown to contribute to sex-limited mimicry in *Papilio* spp. In this report, the authors describe the developmental transcriptome of *P. polytes*; identify several alternative isoforms of *dsx* and quantify their expression in different tissues and at different stages of development using RNA-seq and quantitative PCR; use the developmental transcriptome and motif searches to suggest potential downstream targets of *dsx*; and predict the secondary structure of alternative *dsx* protein isoforms. Although this analysis lacks an experimental or hypothesis-testing component, it provides resources and sets the stage for future experimental analyses, so it is a useful contribution to the field.

In general, the paper is straightforward, and the data are well described. However, I think several issues require clarification or improvement. Some of these relate to potential over-interpretation of the data.

Issues related to correlation network analysis (throughout the paper). This is a major concern of mine. With so few samples, the module structure inferred by this type of analysis is notoriously sensitive to parameter settings. First, the authors need to report their parameters, and show that the modules they infer are at least somewhat robust to parameter settings. Otherwise, the notion

of “dsx-containing module” has little meaning. Second, I strongly suspect that most of the modular structure comes from the use of very different tissues and widely separated developmental stages. In this sense, the different “dsx-containing modules” that the authors report for different tissues/stages may simply reflect differential gene expression between tissues and stages, which mostly has nothing to do with dsx. You have different modules for different tissues/stages, and dsx has to fall out “somewhere” so of course it comes out in different modules in different samples. So perhaps the fact that dsx is associated with different “modules” at different stages/tissues tells you very little about the regulatory relationship between dsx and other genes in these “modules”. The Dsx binding motif is fairly simple, so the fact that many putative “target” genes have that motif may also not mean very much. I urge the authors to re-examine their network analysis more carefully, to understand where the network structure is really coming from, and whether the allocation of dsx to particular modules is robust.

Lines 56-57 “It appears as if this critical gene has evolved multiple mechanisms to maintain and govern different morphs even in closely related species” – This statement, while potentially true, is not directly supported by the data presented in this paper. Frankly, this hypothesis seems to be no more likely after this study than it was before.

dsx expression in unfertilized eggs: can the authors please confirm that the eggs were dissected from ovaries in a way that excluded somatic gonad cells? The detection of dsx transcripts by PCR in eggs from unmated females is surprising. This is PCR – even a low amount of contamination from somatic tissues could potentially account for this result.

Lines 134-150. The description of putative dsx targets seemed quite confusing to me. First, what is the evidence for describing *dvl-3* or *lin* as “known dsx targets”? I don’t know of any direct evidence for that. Second, *Abd-B* is a target of dsx in *Drosophila*; that does not necessarily mean that it’s also a dsx target in *Lepidopterans*. I noticed that *Abd-B* did not show up in the abdominal “module” (Figure 2). Please be more careful in distinguishing confirmed facts from hypotheses.

Lines 176-228. The model at the end of the paper is highly speculative and rests on very little data. Some critical parts of this model are still only conjectures that remain to be tested by experiments. There’s nothing wrong with a light dose of interesting speculation at the end of a paper, but please make clear which parts of the model are more solid, and which are speculative.

Lines 22-223 “Besides *Lepidoptera*, *Coleoptera* is the only other order that has two female-specific exons of dsx”. Actually, the same is true for cockroaches (*Blattella*).

Figure 1 – what exactly are the units for “normalized counts”?

Figure 3 title – you are really describing isoform expression here, not “activity”

Reviewer: 2

Comments to the Author(s)

Manuscript #: RSOS-200792

Title: "Tissue-specific developmental regulation and isoform usage underlie the role of doublesex in sex-limited polymorphic mimicry in *Papilio* swallowtails"

Authors: Riddhi Dexamukh et al.

Comments

doublesex (*dsx*) encodes a transcription factor that generally act as a master regulator for sexual differentiation in insects. Recent studies revealed that *dsx* has diverse pleiotropic effects to govern unique sexual dimorphic traits such as horns in beetles, wings in insects, and mimicry in butterflies.

In this study, the authors focused on the mimetic phenotype especially observed in female wings in *Papilio polytes*. In this species, *dsx* regulates female-limited Batesian mimicry. In an attempt to understand how a critical master regulator *dsx* controls a novel adaptive phenotype just like the Batesian mimicry while maintaining its inherent function of sexual differentiation, the authors performed a developmental transcriptome analysis to identify potential targets of *dsx* through metamorphosis.

Overall, this is a nicely done, and several interesting observations are represented and discussed. But I think that this study lacks several data essential for their conclusion like "Isoforms F2 and F3 contributed to most of the *dsx* expression in mimetic wings" and "elevated expression of *dsx* isoforms F2 and F3 might be sufficient to give rise to mimetic phenotype. Also, it is unclear what of the findings in this study show novelty as compared with the previously reported findings. Therefore, this manuscript is not suitable for publication in this current form.

Major requirements for revision

1. Figure 1A. If the authors want to argue that the higher expression of *dsx* in the forewings and the hindwings is closely related to the mimetic features, then they should compare the expression profile of *dsx* between mimetic females and non-mimetic females. Why did the authors compare it between mimetic females and non-mimetic "males". Such comparison will just simply emerge the difference in *dsx* expression between females and males, which is not directly involved in the mimetic features observed in wings.
2. Figure 2. As pointed above, if the purpose of this study is to identify the putative targets of *dsx* that may be related to the mimetic phenotype, then the authors should compare the transcriptome data between mimetic females and non-mimetic females. The data presented in Figure 2 does not rule out the possibility that it may simply reflect sexual difference because the data was based on the transcriptomic comparison between females and males.
3. Lines 203-207. The authors said that isoforms F2 and F3 contribute to most of *dsx* expression in mimetic wings and that elevated expression of *dsx* isoforms F2 and F3 might be sufficient to give rise to mimetic phenotype. However, again, if the authors want to say so, then they should perform comparative analysis between mimetic females and non-mimetic females.
4. Overall. I think this study severely lacks novelty and uniqueness. It is unclear for me what of the findings in this study show novelty as compared with the previously studies such as Kunte et al (2014), Iijima et al (2018), and Nishikawa et al (2015). The authors should make clear this point.

Author's Response to Decision Letter for (RSOS-200792.R0)

See Appendix A.

RSOS-200792.R1 (Revision)

Review form: Reviewer 2

Is the manuscript scientifically sound in its present form?

Yes

Are the interpretations and conclusions justified by the results?

Yes

Is the language acceptable?

Yes

Do you have any ethical concerns with this paper?

No

Have you any concerns about statistical analyses in this paper?

No

Recommendation?

Accept as is

Comments to the Author(s)

The revised manuscript is now suitable for publication. I am satisfied with the author's responses and plausible explanations.

Decision letter (RSOS-200792.R1)

Dear Dr Kunte,

It is a pleasure to accept your manuscript entitled "Tissue-specific developmental regulation and isoform usage underlie the role of doublesex in sex differentiation and mimicry in *Papilio swallowtails*" in its current form for publication in Royal Society Open Science. The comments of the reviewer(s) who reviewed your manuscript are included at the foot of this letter.

Best regards,

Lianne Parkhouse

Editorial Coordinator

on behalf of Dr Andy Greenfield (Associate Editor) and Steve Brown (Subject Editor)

Reviewer comments to Author:

Reviewer: 2

Comments to the Author(s)

The revised manuscript is now suitable for publication. I am satisfied with the author's responses and plausible explanations.

Appendix A

Dear Editor,

Thank you for considering our manuscript. The comments provided by the Associate Editor and the reviewers have helped in improving the manuscript significantly. We have addressed nearly all the concerns in the revised manuscript (changes marked using Track Changes), and our response to the reviewer comments are below.

ASSOCIATE EDITOR'S COMMENTS:

“In particular, you should address, if possible, the following points: i) ensure that the expression profile comparisons are appropriate; ii) re-examine your network analysis more carefully; iii) make sure there is no over-interpretation of the data and clarify its novelty. If you think you are able to address these concerns, your manuscript will be re-considered by the external reviewers. This is not a provisional guarantee that your manuscript will be found acceptable for publication.”

Response: We have now clarified the original contribution of our work and the advances that we offer to the field in the last two paragraphs of Introduction, and in the last paragraph of Results and Discussion. We have addressed the remaining points below in our responses to reviewer comments.

REVIEWER: 1

“Issues related to correlation network analysis (throughout the paper). This is a major concern of mine. With so few samples, the module structure inferred by this type of analysis is notoriously sensitive to parameter settings. First, the authors need to report their parameters, and show that the modules they infer are at least somewhat robust to parameter settings. Otherwise, the notion of “dsx-containing module” has little meaning. Second, I strongly suspect that most of the modular structure comes from the use of very different tissues and widely separated developmental stages. In this sense, the different “dsx-containing modules” that the authors report for different tissues/stages may simply reflect differential gene expression between tissues and stages, which mostly has nothing to do with dsx. You have different modules for different tissues/stages, and dsx has to fall out “somewhere” so of course it comes out in different modules in different samples. So perhaps the fact that dsx is associated with different “modules” at different stages/tissues tells you very little about the regulatory relationship between dsx and other genes in these “modules”. The Dsx binding motif is fairly simple, so the fact that many putative “target” genes have that motif may also not mean very much. I urge the authors to re-examine their network analysis more carefully, to understand where the network structure is really coming from, and whether the allocation of dsx to particular modules is robust.”

Response: Our co-expression analysis with WGCNA followed the standard protocol (also provided as a tutorial with the R package), mostly using default parameters in addition to others that were calculated as a part of this analysis. In systems where we cannot predict how the data might behave, the authors of the WGCNA package recommend using default parameters as they work well across a wide range of experiments. We have explained some of these steps in the methods section (lines 101-105). We separated data for the wing and abdominal tissues across stages prior to the co-expression analysis and performed a WGCNA run for each combination of phenotype and tissue. We compared the co-expressed genes after completing the four runs. While we agree there would be some effect of the stage on the expression of genes, the tissues were the same in each case. We re-checked expression patterns of all the genes reported in Fig. 2 and their correlation with *dsx* expression in the respective tissues (the correlation coefficients for co-expressed genes in each comparison have been added to supplementary Table S4), and modified the table in Fig. 2 as well. Most of the genes represented in that table now show strong correlations with *dsx* expression (>0.75 , Pearson correlation coefficient). The genes relevant in mimetic wings show a peak in 3-day pupal wings compared to other stages. Non-mimetic wings and abdomen had low expression of *dsx* to begin with and the co-expression profiles here may not mean much, as we discuss in the paper (lines 171-173). However, some genes that came up as relevant in the non-mimetic wings showed wing-specific expression irrespective of sex, which we have

retained, as they may perform a more generic function related to wing development, we have also mentioned this in the main text (lines 173-175).

“Lines 56-57 “It appears as if this critical gene has evolved multiple mechanisms to maintain and govern different morphs even in closely related species” – This statement, while potentially true, is not directly supported by the data presented in this paper. Frankly, this hypothesis seems to be no more likely after this study than it was before.”

Response: This statement mainly referred to the genetic basis of mimicry in *Papilio memnon* and *Papilio polytes*. While *dsx* regulates mimicry in both these species, its genetic architecture differs in the closely related species and this may reflect in its molecular mechanism of developmental regulation as well. Our intention was to imply that close examination of *dsx* activity at the molecular level might help us understand how this gene regulates such diverse phenotypes in different species. We have clarified this statement in the main text (lines 57-66).

dsx expression in unfertilized eggs: can the authors please confirm that the eggs were dissected from ovaries in a way that excluded somatic gonad cells? The detection of *dsx* transcripts by PCR in eggs from unmated females is surprising. This is PCR – even a low amount of contamination from somatic tissues could potentially account for this result.

Response: While sampling the unfertilized eggs from unmated females, we tried our best to remove all the tissue from around the eggs as closely as possible. Have now mentioned this in methods in lines 81-82.

Lines 134-150. The description of putative *dsx* targets seemed quite confusing to me. First, what is the evidence for describing *dvl-3* or *lin* as “known *dsx* targets”? I don’t know of any direct evidence for that. Second, *Abd-B* is a target of *dsx* in *Drosophila*; that does not necessarily mean that it’s also a *dsx* target in *Lepidopterans*. I noticed that *Abd-B* did not show up in the abdominal “module” (Figure 2). Please be more careful in distinguishing confirmed facts from hypotheses.

Response: We apologize for the lack of clarity in that section. We were referring to the relevance of *Abd-B* and Wnt pathway to *dsx* activity and that our results showed links to these pathways. We rigorously scrutinized all our WGCNA hits and their correlations with *dsx* expression. Our re-examination of WGCNA results cast some doubt on *lin* as a suitable candidate because despite its wing-specific expression, it did not show a peak in 3-day pupal stage similar to *dsx*. We have modified the text accordingly. At the same time, we now highlight *osa-like* as an important candidate that showed female-biased expression in 3-day pupae and high correlation with *dsx* expression in wings, and which governs genes involved in wing patterning. This gene was earlier included in Fig. 2 as a key candidate, but we had not highlighted it in the text. We have modified lines 160-166 to accommodate this. We also acknowledge that ChIP-Seq and co-IP would be the best way forward to identify physical targets of *dsx*. The absence of *Abd-B* and *Abd-A* in the abdomen samples indicates that the expression of these two genes was not correlated with that of *dsx* in the abdomen. They might have clustered with other abdomen-specific genes in a separate module.

“Lines 176-228. The model at the end of the paper is highly speculative and rests on very little data. Some critical parts of this model are still only conjectures that remain to be tested by experiments. There’s nothing wrong with a light dose of interesting speculation at the end of a paper, but please make clear which parts of the model are more solid, and which are speculative.”

Response: We have added lines 217-222, and modified other parts of this section along with the figure legend of Fig 4 to clarify which aspects of panels 4A and 4B are based on our results and which aspects of the model need to be tested further.

Lines 22-223 “Besides *Lepidoptera*, *Coleoptera* is the only other order that has two female-specific exons of *dsx*”. Actually, the same is true for cockroaches (*Blattella*).

Response: Thank you for bringing this to our notice. We have modified that statement in lines 277-279 and added a citation for the same.

Figure 1 – what exactly are the units for “normalized counts”?

Response: Normalized counts refer to the number of reads aligning to a gene after accounting for library size and composition for all the samples in the dataset. Due to this, normalized counts do not have units. We believe that this is a common practice.

Figure 3 title – you are really describing isoform expression here, not “activity”

Response: Thank you for bringing this to our notice. We have modified the title. We have also added an updated version of Fig. 3 in the revision.

REVIEWER: 2

“... But I think that this study lacks several data essential for their conclusion like "Isoforms F2 and F3 contributed to most of the *dsx* expression in mimetic wings" and "elevated expression of *dsx* isoforms F2 and F3 might be sufficient to give rise to mimetic phenotype. Also, it is unclear what of the findings in this study show novelty as compared with the previously reported findings. Therefore, this manuscript is not suitable for publication in this current form.

1. Figure 1A. If the authors want to argue that the higher expression of *dsx* in the forewings and the hindwings is closely related to the mimetic features, then they should compare the expression profile of *dsx* between mimetic females and non-mimetic females. Why did the authors compare it between mimetic females and non-mimetic "males". Such comparison will just simply emerges the difference in *dsx* expression between females and males, which is not directly involved in the mimetic features observed in wings.”

Response: Males and non-mimetic females have similar wing pattern and phenotype, therefore male or female-specific *dsx* isoforms are not altering the non-mimetic wing pattern in a sex-specific manner. While previous work in Kunte et. al., 2014 and Nishikawa et. al., 2015 have compared *dsx* expression in mimetic and non-mimetic females, we do not fully understand the role of *dsx*, if any, in wing patterning in non-mimetic individuals. Our motivation behind the use of males (instead of non-mimetic females) was that it might help us understand the role of *dsx* in wing development in the absence of mimicry and we could compare genes between males and mimetic females to screen mimicry-specific candidates. At the same time, we were also unable to obtain samples for non-mimetic females across developmental stages, despite several attempts to establish pure-breeding non-mimetic lines in the lab during sampling for this work (this form is relatively uncommon in India).

2. Figure 2. As pointed above, if the purpose of this study is to identify the putative targets of *dsx* that may be related to the mimetic phenotype, then the authors should compare the transcriptome data between mimetic females and non-mimetic females. The data presented in Figure 2 does not rule out the possibility that it may simply reflect sexual difference because the data was based on the transcriptomic comparison between females and males.

Response: We agree that the data in Figure 2 might represent some sex-specific candidates, however, since males and non-mimetic females share wing phenotypes and possibly the underlying genetic network responsible for those phenotypes. It is still a useful comparison to make to find phenotype-specific wing development candidates irrespective of sex. However, we have modified relevant text to reflect this limitation.

3. Lines 203-207. The authors said that isoforms F2 and F3 contribute to most of *dsx* expression in mimetic wings and that elevated expression of *dsx* isoforms F2 and F3 might be sufficient to give rise to mimetic phenotype. However, again, if the authors want to say so, then they should perform comparative analysis between mimetic females and non-mimetic females.

Response: We were able to compare isoforms between mimetic and non-mimetic females at 3-day pupal stage and observed the same results, with very little expression of F1 and mostly F2 and F3 contributing to *dsx* expression. However, since we were unable to obtain samples for non-mimetic females for other stages, we did not show this data previously. We have now modified Fig. 3 and added a panel showing this comparison. We have also modified that statement to avoid drawing firm conclusions solely based on qPCR and expression data.

4. Overall. I think this study severely lacks novelty and uniqueness. It is unclear for me what of the findings in this study show novelty as compared with the previously studies such as Kunte et al (2014), Iijima et al (2018), and Nishikawa et al (2015). The authors should make clear this point.

Response: Most of the work in this manuscript builds on previously published work from these three studies, however, here we have tried to compare the role of *dsx* in regulating wing patterns in mimetic butterflies (novel function in *Papilio polytes*) and sex-specific development (conserved function of *dsx* across insects), which has not been done before. We have tried to understand the molecular nature of *dsx*-based developmental regulation by comparing its potential targets and examining the expression of its isoforms in the two sets of tissues and functions that *dsx* performs. We have added lines 57-76 to clarify this.